# The Vascular Side of Chronic Bed Rest: When a Therapeutic Approach Becomes Deleterious

**DOI:** 10.3390/jcm9040918

**Published:** 2020-03-27

**Authors:** Anna Pedrinolla, Alessandro L. Colosio, Roberta Magliozzi, Elisa Danese, Emine Kirmizi, Stefania Rossi, Silvia Pogliaghi, Massimiliano Calabrese, Matteo Gelati, Ettore Muti, Emiliano Cè, Stefano Longo, Fabio Esposito, Giuseppe Lippi, Federico Schena, Massimo Venturelli

**Affiliations:** 1Department of Neuroscience, Biomedicine, and Movement Science, Section of Movement Science, University of Verona, 37134 Verona, Italy; anna.pedrinolla@univr.it (A.P.); alessandro.colosio@univr.it (A.L.C.); silvia.pogliaghi@univr.it (S.P.); federico.schena@univr.it (F.S.); 2Department of Neurological and Movement Sciences, Section of Neurology, University of Verona, 37134 Verona, Italy; roberta.magliozzi@univr.it (R.M.); stefania.rossi@univr.it (S.R.); calabresem@hotmail.it (M.C.); 3Department of Life and Reproduction Sciences, Laboratory of Clinical Biochemistry, University of Verona, 37134 Verona, Italy; elisa.danese@univr.it (E.D.); matteo.gelati@univr.it (M.G.); giuseppe.lippi@univr.it (G.L.); 4Department of Physiology, Faculty of Medicine, Uludag University. Eskisehir City Hospital, Eskisehir 16059, Turkey; kirmiziemine@gmail.com; 5Mons. Mazzali Foundation, 46100 Mantua, Italy; Ettore.muti@fondazionemazzali.it; 6Department of Biomedical Sciences for Health, University of Milan, 20133 Milan, Italy; emiliano.ce@unimi.it (E.C.); stefano.longo@unimi.it (S.L.); fabio.esposito@unimi.it (F.E.); 7IRCCS Galeazzi Orthopedic Institute, 20133 Milan, Italy; 8Department of Internal Medicine section of Geriatrics, University of Utah, Salt Lake City, UT 84132, USA

**Keywords:** Vascular function, Inflammatory profile, physical constrain, bed-rest

## Abstract

The interplay between chronic constraint and advanced aging on blood flow, shear-rate, vascular function, nitric oxide (NO)-bioavailability, microcirculation, and vascular inflammation factors is still a matter of debate. Ninety-eight individuals (Young, *n* = 28, 23 ± 3 yrs; Old, *n* = 36, 85 ± 7 yrs; Bedridden, *n* = 34, 88 ± 6 yrs) were included in the study. The bedridden group included old individuals chronically confined to bed (3.8 ± 2.3 yrs). A blood sample was collected and analyzed for plasma nitrate, and vascular inflammatory markers. Hyperemic response (∆peak) during the single passive leg movement (sPLM) test was used to measure vascular function. Skeletal muscle total hemoglobin was measured at the vastus lateralis during the sPLM test, by means of near infrared spectroscopy (NIRS). Bedridden subjects revealed a depletion of plasma nitrates compared with Old (−23.8%) and Young (−31.1%). Blood flow was lower in the Bedridden in comparison to Old (−20.1%) and Young (−31.7%). Bedridden presented lower sPLM ∆peak compared Old (−72.5%) and the Young (−83.3%). ∆peak of NIRS total hemoglobin was lower in the Bedridden compared to that in the Young (−133%). All vascular inflammatory markers except IL-6 were significantly worse in the Bedridden compared to Old and Young. No differences were found between the Old and Young in inflammatory markers. Results of this study confirm that chronic physical constraint induces an exacerbation of vascular disfunction and differential regulation of vascular-related inflammatory markers. The mechanisms involved in these negative adaptations seems to be associated with endothelial dysfunction and consequent diminished NO-bioavailability likely caused by the reduced shear-rate consequential to long-term reduction of physical activity.

## 1. Introduction

The state of the vascular system is one of the most important factors in determining health during aging because it transports oxygen and nutrients throughout the body and, therefore, dictates when and how organs and systems will suffer and ultimately fail [1]. Although chances to age healthily exist, as the time goes by, the vascular system naturally brings with itself several biochemical, enzymatic, and cellular changes leading to modification of the signals that modulate it, conferring enormous risk for the development of cardiovascular disease and determining its threshold and severity once it is manifested [2]. Indeed, several concatenated events develop just due to the time flow: a shift towards a pro-inflammatory vascular phenotype with upregulation of inflammatory cytokines, chemokines, and adhesion molecules in the vascular walls, and a reduction of endothelial function, which in turn affect circulation [1,3,4]. In light of this, and considering that everyone’s watch runs, every human being of any age should make a hefty effort to protect their vascular system and preserve it while aging. Physical activity is crucial for the integrity of the vascular system and limiting activity over days, weeks, or even years may lead to problems in almost every major organ [5]. Indeed, during periods of inactivity, some physiological processes might become underused with the consequent development of dysfunctions at several physiological levels [6]. At the vascular level, the intermediate mechanism involved in the deleterious effect of lack of mobility might mainly dwell in the endothelium, which plays a key role in arterial function through synthesis and release of biologically active molecules that can influence its own function in an autocrine or paracrine fashion [3]. Already, during healthy aging, the capacity of the vascular endothelium to generate and use one of its derived relaxing factors, nitric oxide (NO), declines [7]. In addition to the activity of autonomic nervous system that regulates vasomotor function at the systemic level, a peripheral stimulus for the release of NO from endothelial cells is augmented blood flow or, more specifically, the frictional force along endothelial cell membranes, also known as shear-stress [8]. When mobility is dramatically limited, the reduction of shear-stress produced from the dramatic reduction of movement-induced increase in blood flow removes the stimulus for NO biosynthesis with consequent reduction of vasodilation (acute) and structural enlargement (chronic) adaptations [6]. Therefore, lack of activity with reduction of shear-stress indirectly modulates specific metabolites needed for vascular health. This framework comes into play in a common therapeutic approach still widely used in hospitals and nursing homes, which consists in the full bed rest of elderly individuals in order to better manage chronic conditions [9,10]. Not surprisingly, it has been seen that bedridden elderly people more easily develop morbidity including faster development of cardiovascular disease compared with physically active elderly people [10,11]. However, the majority of the studies aiming to investigate the physiological effects of bed rest have included healthy adults, and the induced bed-rest periods investigated varied from few days [12] to few weeks [13] making it difficult to translate these findings directly to critically ill elderly individuals. Indeed, these models induce a sudden and extreme local and/or systemic physical constrain and do not reflect typical human sedentary behavior that may be associated with more gradual and inexorable effects. Furthermore, the effect of chronic bed rest on NO-mediated endothelial function and potential inflammatory effects is still matter of a debate. Thus, the aim of this study was to investigate potential changes of NO-bioavailability, circulation, and microcirculation and levels of serum inflammatory biomarkers in a group of chronic bedridden, oldest–old, residing in a long-term facility compared with a group of active, age-matched counterparts. In order to consider the effect of aging, we compared the active and bedridden elderly with a group of young adults. Our working hypothesis was that bedridden individuals would show poorer NO-bioavailability, accompanied by a poorer systemic circulation and higher inflammatory status compared with active individuals. 

## 2. Methods

### 2.1. Participants 

Chronic bedridden people (3.8 ± 2.3 years) oldest–old, 80 years old and older were recruited at the Geriatric Institute Mons. Arrigo Mazzali Foundation (Mantua, Italy). Age-matched active oldest–old participants and young adults were recruited from the same geographical area. Individuals were excluded from the study in the presence of neurodegenerative disease (i.e., Parkinson’s disease, Alzheimer’s disease); heart, liver, or kidney failure; organ transplantation; hemorrhage; neuromuscular disease; or any other conditions limiting the assessment procedures. All experiments were conducted after informed and written consent was obtained from the subjects and their relatives and elderly and younger participants in accordance with the Declaration of Helsinki, as part of a protocol approved by the Institutional Review Board of the Department of Neurosciences, Biomedicine, and Movement Sciences, University of Verona, Italy (Verona, Italy - #CT241123; NIH Clinical trial identification number: NCT03087643).

### 2.2. Study Overview

All assessment procedures were performed in the morning between 9.00 and 12.00 a.m. When subjects included in the study reached the ambulatory at the Geriatric Institute Mons. Arrigo Mazzali Foundation, first a blood sample was collected by an expert nurse working at the foundation, and anthropometric measures for the determination of thight volume of both right and left limbs were executed [14]. Thight volume was calculated based on thigh circumferences (three sites: distal, middle, and proximal), thigh length, and skinfold measurements using the following formula:Thigh Volume = (L/12_Π_)·(C1^2^ + C2^2^ + C3^2^) − [(S − 0.4)/2]·L·[C1 + C2 + C3)]/3]
where L refers to the length; C1, C2, and C3 refer to the proximal, middle, and distal circumferences, respectively; and S is skinfold thickness of the thigh. 

Following the blood sample collection, participants were placed in an upright-seated position, near-infrared spectroscopy (NIRS) probes were set on both vastus lateralis and subjects were left at rest in this position for 10 minutes. Subsequently, the single passive leg movement (sPLM) test was performed on one leg, and after 10 minutes rest the test was performed on the other leg [15]. All measurements taken in both right and left limbs were then averaged. 

### 2.3. Blood Sample Collection 

Venous peripheral blood (18 mL) was collected between 9:00 and 10:00 from bedridden individuals, active elderly, and younger adults in a fasted state. Within 45 minutes, plasma was separated from peripheral blood by centrifugation (1200 rpm for 20 min at 4 °C) and kept at −80 °C until analysis. Plasma samples were ultrafiltrated through a 30 kDa molecular weight cutoff filter (cat. no. UFC503096) (Millipore, Molsheim, France) to reduce background absorbance.

### 2.4. NO Bioavailability via Plasma Nitrates Assessments 

Nitrate concentration was detected by nitrate/nitrite colorimetric assay kit (cat. no. 780001) (Cayman Chemical Co., Ann Arbor, MI, USA) according to the manufacturer’s protocol. The detection limit of nitrate was 2.5 µM. The nitrate concentration was analyzed in duplicate and read against the manufacturer’s standard curve.

### 2.5. NO-Bioavailability via sPLM 

Recent investigations have revealed that sPLM-induced hyperemia is predominantly a consequence of NO-mediated vasodilation [15]. Therefore, we have adopted this noninvasive and reliable method to determine NO bioavailability [15,16]. During the test, the subjects rested in the upright-seated position for 10 min before the start of data collection and remained in this position throughout this part of the study. The sPLM protocol consisted of 30 s of resting baseline femoral blood flow data collection, followed by one single passive knee flexion and extension with the same measure for the following 60 s. sPLM was performed by a member of the research team, who moved the subject’s lower leg through a 90° range of motion (180°–90° knee joint angle). Blood mean velocity (V_mean_) was analyzed with 1 Hz resolution on the Doppler ultrasound system (GE Logiq-7) for 30 s at rest and second by second for the 60 s following the single passive movement. Resting arterial diameter, resting blood flow, resting shear-rate, relative changes (∆peak) from rest, and area under the curve (AUC) of femoral blood flow were determined for each subject. Arterial diameter was measured as the distance (mm) between the intima–lumen interfaces for the anterior and posterior walls in the common femoral artery. Blood flow and shear-rate were calculated using arterial diameter blood velocity according to these formulae [16]: (a) Blood Flow (mL/min) = V_mean_ × ᴨ × (vessel diameter/2)^2^ × 60,
(b) Shear-Rate (s − 1) = 8V_mean_/vessel diameter.

Blood flow and AUC were normalized for the volume of the thigh [14].

### 2.6. Immunoassay Protein Analysis 

The serum level of a pattern of inflammatory mediators potentially involved in vascular pathological alterations, including tumor-necrosis factor-α (TNF-α), interleukin (IL)-1β, IL-6, IL-8, interferon-γ (IFN-γ), platelet-derived growth factor (PD-GF), granulocyte-macrophage colony stimulating factor (GM-CSF), and the chemokine (C–C motif) ligand 5 (CCL5), also named regulated on activation, normal T cells expressed and secreted (RANTES), were assessed using a combination of immune-assay multiplex techniques based on the Luminex technology (27-Plex, Bio-Plex X200 System equipped with a magnetic workstation, Bio-Rad, Hercules, CA, USA) previously optimized [17]. All samples were run in duplicate in the same experiment and in two consecutive experiments, in order to verify the reproducibility and consistency of the results. 

### 2.7. Total Hemoglobin in Microcirculation Via Near-infrared Spectroscopy During sPLM Test 

The total hemoglobin in microcirculation was evaluated as previously described [18] using a quantitative near-infrared spectroscopy (NIRS) system (Oxiplex TSTM, ISS, Champaign, USA) that provided continuous measurement (sampling frequency 1 Hz) of absolute concentrations (µM) of total hemoglobin. After shaving, cleaning, and drying of the skin area, the NIRS probe was positioned longitudinally on the belly of the vastus lateralis muscle ~15 cm above the patella, attached to the skin with a bi-adhesive tape and secured with elastic bandages around the thigh. The device was calibrated before each test after a warm-up of at least 30 minutes as per manufacturer recommendations. Baseline value was calculated as mean of the 20 s before sPLM. Then, the NIRS signal was treated by subtracting the baseline value and the following indexes were calculated to evaluate the response after the sPLM: maximal value reached after the knee flexion–extension (∆peak) as indicative of the magnitude of the response to the PLM; and AUC using the same data analysis procedure utilized for the ultrasound data.

### 2.8. Statistical Analysis 

All statistical analysis was performed with SigmaPLOT Windows Version 14.0 (Systat Software, Chicago, IL, USA). In consideration of preliminary results on peripheral circulation values and vascular function measurements, a sample size of 32 participants for each group was selected to ensure a statistical power higher than 0.80 and a type 1 error <0.05. First, normality was assessed by Shapiro–Wilk test. Subsequently, Kruskal–Wallis one-way analysis of variance on ranks was performed to assess between groups differences, followed by the Tukey test. For detecting between-groups differences of blood flow and total hemoglobin during the PLM test, a one-way repeated-measures analysis of variance on ranks was performed, followed by the Tukey test. α level was set at 0.05. If not differently stated, data are presented as mean ± SD.

## 3. Results

### 3.1. Subjects Characteristics

A total of 98 individuals (Young = 28, Old = 36, Bedridden = 34) were included in the study. The reason why individuals were bedridden was mostly related to age and age-related frailty or weakness with consequent motor dysfunction, and in some case elevated risk of fall. Significant differences were found for all variables, except for weight, height, and BMI in the Young group compared with Old and Bedridden groups. Significant differences between the Old and Bedridden were detected for the following variables: thigh volume (*p* = 0.041), number of comorbidities (*p* = 0.002), cardiovascular disease (*p* = 0.032), number of medications (*p* = 0.039), antipsychotics (*p* < 0.001), antidepressant (*p* < 0.001), and benzodiazepines (*p* = 0.037). Table 1 shows the characteristics of the three groups in detail. As shown in Table 1, the most frequent comorbidities in the Bedridden group were cardiovascular disease and arthritis followed by diabetes. Diabetes and arthrosis were common comorbidities in the Old group too. The most frequently used medications by the Bedridden group were antihypertensives, antidepressants, and benzodiazepines. While in the Old group the most frequently used medications were antihypertensives (Table 1). 

### 3.2. NO-bioavailability 

Significant reduction of plasma nitrates has been found in the Bedridden with respect to the Old (*p* = 0.041) and Young (*p* < 0.001) (Table 2). Furthermore, the blood flow ∆peak in response to sPLM was found reduced in the Bedridden with respect to that in the Old (*p* = 0.020) and Young (*p* < 0.001), and in the Old with respect to that in the Young (*p* < 0.001) (Table 2). Furthermore, blood flow AUC relative to sPLM was found to be reduced in the Bedridden compared to that in the Young (*p* < 0.001), and in the Old compared to that in the Young (*p* < 0.001) (Table 2, Figure 1). Concerning ∆peak, the AUC was found to be reduced exclusively in the Bedridden compared to that of the Young (*p* < 0.001; *p* < 0.001, respectively) (Table 2, Figure 1). 

### 3.3. Circulation and Microcirculation 

Significant reduction of resting blood flow and resting shear-rate were found between the Bedridden and Young (*p* < 0.002; *p* < 0.001, respectively) and the Old (*p* > 0.001; *p* = 0.002, respectively). Difference in resting total hemoglobin were found only between the Bedridden and Young only (*p* < 0.001; Table 2, Figure 1). 

### 3.4. Serum Inflammatory Profiles

Significant reduction of IL-8 (*p* = 0.011), TNF-α (*p* = 0.009), RANTES (*p* < 0.001), and PDGF-b (*p* < 0.001) has been in found in the Bedridden with respect to the Old (Table 2, Figure 2). Furthermore, TNF-α (*p* = 0.048), RANTES (*p* = 0.004), and PDGF-b (*p* = 0.034) levels were found reduced in the Bedridden with respect to those in the Young (Table 2, Figure 2). On the contrary, IL-1β (*p* = 0.013, *p* = 0.004) and GM-CSF (*p* = 0.007, *p* < 0.001) were found increased in the serum of the Bedridden with respect to both the Old and the Young (Table 2, Figure 2), while IFN-γ (*p* = 0.007) levels were found increased in the serum of the Bedridden only compared to that of the Young (Table 2, Figure 2). Interestingly, no differences in serum levels of IL-6 were found among the three examined groups (Table 2, Figure 2).

## 4. Discussion

Although the effect of physical constrain and bed rest on vascular function has been already investigated, to our knowledge, this is the first study measuring vascular function and its related inflammatory profile in the chronically bedridden oldest–old. In the present study, we assessed NO-bioavailability via plasma NO-metabolites and sPLM induced hyperemia, circulation, and microcirculation, as well as the inflammatory profile in chronically bedridden oldest–old individuals and compared to a group of age-matched physically active oldest–old and a group of young active adults. The main finding of this study was that NO-bioavailability, circulation, microcirculation, and inflammatory profile seem to be severely altered by the chronic physical constraint due to years (3.8 ± 2.3 years) of bed rest rather than due to the aging process itself. Indeed, all variables were found to be significantly deteriorated in the Bedridden compared to the Old and Young, but no differences were detected between those latter two groups, except for blood flow ∆peak and AUC, where the aging effect appeared. According to our hypothesis, these data suggest a deleterious effect of physical constraint on vascular function, and this appears to be mediated by endothelial dysfunction with reduced NO-bioavailability, consequent poor circulation, and systemic inflammation. The mechanisms supporting this mal-adaptation of the vascular system might be triggered by the absence of physical activity, which does not provide the stimulus, shear-rate, to produce metabolites indispensable for vascular health, such as NO.

### 4.1. Evidence That Physical Constraint Affects NO-Bioavailability

Nitric oxide, an unstable free radical endogenously produced by several cell types, exerts essential biological regulatory functions in the whole vascular system, including arteries and capillaries [19]. For this reason, a depletion of NO-bioavailability is one of the mechanisms in the pathogenesis of endothelial dysfunction affecting vasculature, increasing the risk of developing cardiovascular disease and related comorbidities [19,20,21]. Nyberg et al. [22] measured NO-bioavailability via plasmatic and muscular nitrates in young, older life-long sedentary, and older life-long physically active individuals. Results demonstrated that NO-bioavailability was compromised at both systemic and muscular levels of sedentary aging humans. Interestingly, life-long physically active older participants did not show the same trend compared to young individuals [22]. Moreover, Nosova et al. [23] evaluated vascular function in healthy adults, after only five days of bed rest. Results of this study showed a significant reduction in the NO-mediated hyperemic response, measured at the brachial and femoral arteries by means of flow-mediated dilation, supporting the idea that NO-mediated endothelial dysfunction occurs in a very short time in response to inactivity [23]. Our data are in agreement with those previous findings, showing a drop in NO-bioavailability based on the reduction of plasma nitrates (Table 2) in chronically physically constrained individuals compared with active oldest–old and younger controls, further supported by a reduction of sPLM-induced hyperemic response (Table 2, Figure 1), confirming the pivotal role of chronic inactivity on NO-mediated endothelial dysfunction. 

Moreover, this is the first study applying NIRS during the sPLM test for measuring NO-mediated response in muscular microcirculation. However, previous studies have examined the endothelium-dependent dilation in skin microcirculation using a laser Doppler [13,24]. Demiot et al. [13] measured cutaneous microcirculatory endothelial-dependent vasodilation in sedentary and active healthy adults before and after 56 days of bed rest. Results showed that in sedentary individuals, endothelium-dependent vasodilation in microcirculation after the bed-rest period was significantly reduced, while it was preserved in active subjects [20]. Furthermore, impaired skin microcirculation was also found in the inactive limbs of spinal cord injury individuals, which reflects in part a model of chronic disuse [25,26]. Although cutaneous and skeletal muscle microcirculation might differ in response to stimulus and stress, the results of Demiot et al. [13] support our findings. Indeed, in our study, total hemoglobin measured by NIRS during the sPLM test, serving as an indicator of muscular microcirculation, showed a different response in chronically physically constrained subjects compared with active counterparts and younger individuals (Table 2, Figure 1). Specifically, young participants exhibited a reduction of total hemoglobin few seconds after the single passive movement followed by a fast increase, over the baseline values (Figure 1, panels A and D). The active oldest–old exhibited the same pattern, but the rise of total hemoglobin following the drop was reduced compared with that of younger subjects, reaching basal values without exceeding them (Figure 1, panels B and E). Even more interestingly, chronically physically constrained individuals exhibited a more important drop followed by a slow rise of total hemoglobin, and the basal values were not reached in 60 seconds (Figure 1, panels C and F). Although the microcirculatory response to sPLM has not been investigated previously and more studies for understanding its pattern and the mechanisms are needed, our results lead us to speculate that microcirculatory endothelial-dysfunction participates in vascular deconditioning in response to chronic physical constraint.

### 4.2. Evidence That Chronic Physical Constraint Affects Circulation and Microcirculation

Bleeker et al. [27] investigated the effect of 52 days of bed rest on arterial dimension and circulation in healthy men. Results of this study showed a reduction of arterial diameter not coupled with a reduction of rest blood flow at the femoral artery following several days of inactivity [27]. In accordance, Nyberg et al. [22] reported no difference in resting femoral artery blood flow in life-long sedentary older individuals compared with their active counterparts and young subjects. Our results are not in agreement with these previous studies, showing a reduction of femoral artery rest blood flow in chronically physically constrained individuals compared to their active counterparts and young subjects (Table 2, Figure 1). The reason that previous evidence does not support our results might be due to the characteristics of subjects included in these studies. Indeed, in the article by Bleeker at al. [27], the 52-day of bed rest in healthy adult subjects, was probably not enough for inducing structural adaptations. In the paper by Nyberg et al. [22], the life-long sedentary subjects consisted of “elderly with less than 2 hours of moderate intensity exercise per week during the last 30 years”, consequently they were sedentary, but they still walked and probably had a regular daily life. While subjects included in our study were the chronically bedridden oldest–old, and the total inactivity during the last years might have induced a structural adaptation of the inactive limbs with consequent reduction of blood flow. 

Furthermore, microcirculation has been seen to be affected by inactivity as well. Indeed, microcirculation mainly serves as an exchange site for nutrients between blood and the surrounding tissue and the lowered demand of nutrients by muscles and other organs in a state of chronic inactivity imposes functional and structural changes at the microvascular level as well [24]. Demiot et al. [13] revealed a significant reduction in resting cutaneous microcirculation, measured by means of laser Doppler, after 56 days of bed rest in sedentary individuals. Furthermore, Zafeiridis et al. [28] measured resting muscle perfusion and capillary flow in vastus lateralis muscle of spinal cord-injured and able-bodied individuals by means of local clearance of a radioactive tracer (^99m^Tc-pertechnetate). Results showed that microcirculation in paralyzed subjects was reduced by ~40% compared with that in able-bodied subjects [28]. Again, although the methods and the subjects included in the previous studies are different from ours, the results are in line with our findings. Indeed, chronic bedridden subjects exhibited reduced total hemoglobin, an indicator of muscular microcirculation, measured at rest in vastus lateralis muscle compared with active-counterparts and younger subjects. These evidences suggest that chronically reduced blood flow due to inactivity, as in the case of spinal cord injured and chronic bedridden individuals, may impair vessel structure and function, increasing vascular resistance, reducing capillary number and diameter [13,24,28].

### 4.3. Evidence That Physical Constraint Affects Vascular-Related Inflammatory Profile

It is known that bed rest activates metabolic and inflammatory markers [29], however a comprehensive analysis of the effect of physical inactivity and advanced aging on inflammatory serum profile was never achieved. By analyzing serum samples, we detected significant changes in the inflammatory profile of the examined subjects. In particular, we observed the reduction of IL-8, TNF-α, RANTES, and PDGF-b in Bedridden with respect to Old individuals, and increased serum levels of IL-1β and GM-CSF in the Bedridden with respect to both the Old and Young, as well as the increased levels of IFN-γ in the serum of the Bedridden only compared to that of the Young. All these data support a complex alteration and/or activation of pro- and anti-inflammatory responses due to inactivity, which was previously partially explained as “inflamm-aging” able to induce a hyper-inflammatory state commonly found associated with aging [30]. These findings suggest that a complex regulation of inflammatory reactions may occur in the Bedridden condition, possibly reflecting a complex modulation of innate and adaptive immune responses, and requires further and more in depth cell and molecular studies.

It is important to mention the useful role of serum protein analysis in order to identify potential biomarkers of disease outcome and inflammatory conditions by using standard laboratory procedures with final diagnostic and prognostic values. Most of the observed cytokine alterations in the serum appear to be mainly linked to innate immune response, in particular to monocyte/macrophage activation, suggesting that physical constraint and/or advanced aging have a relevant effect on regulating the balance of innate immune activity.

### 4.4. Physiological Considerations on Aging and Physical Constraint

The mechanism that triggers the mal-adaptation of vascular function in response to chronic bed rest seems to have as a main protagonist the endothelium. Under normal conditions, the endothelium induces vasodilation, limits vascular inflammation, and maintain blood fluidity [13]. Already during the normal aging process, a modification of the vascular system occurs leading to changes in the signals that modulate vascular responses, supporting the development of cardiovascular disease. However, when the absence of activity is added to the natural aging process, and inactivity becomes chronic, the diminished blood flow in response to the diminished metabolic demand due to diminished movement, leads to significant decrement of the most important stimulus for the endothelial function: the shear-rate (Figure 3). The decline in frequency and intensity of this signal due to the physical constraint contributes to physiological inactivity-mediated endothelial dysfunction [13,24,31]. Although the Bedridden showed significantly poorer vascular outcomes compared to the Old, we cannot totally exclude that medications taken by bedridden individuals might have had an additive role in the worsening of their vascular function.

### 4.5. The Importance of Shear-Rate during Aging and Physical Constraint

In response to shear-rare, the endothelium produces and releases vasoactive substances, such as NO, playing a central role in local control of arteries and capillaries size and consequently tissue perfusion, which is why the production of NO in endothelial cells is considered one of the most important vasodilator mechanism responsible for the preservation of vascular structure and function [21,32]. However, when the shear-rate is strongly diminished, NO production and utilization is reduced as well. Thus, reduced NO-bioavailability results in major detrimental alterations of vascular function, including vasoconstriction and capillary atrophy [21] (Figure 3). 

Decreased NO bioavailability also promotes proliferation of smooth muscle cells, platelet aggregation, white blood cell adhesion, supporting a pro-inflammatory status, confirming its key role in the initiation and progression of vascular disease [21] (Figure 3). Thus, these facts lead to the conclusion that inactivity-induced endothelial dysfunction, with the reduction of NO-bioavailability in arteries and capillaries and promoting a vascular inflammatory profile, increases the risk of developing cardiovascular disease and related comorbidities (Figure 3). 

## 5. Conclusions

The results of this study confirm previous findings about the deleterious effect of physical constraint on vascular function. However, to our knowledge this is the first study that investigates endothelial-mediated vascular function and NO-bioavailability in chronic bedridden oldest–old, giving evidence about the gradual and inexorable effect of this condition without inducing a sudden and extreme inactivity in healthy individuals. Interestingly, the link between chronic physical constraint and the detrimental alteration of vascular function seems to be the endothelial dysfunction caused by the diminished (in frequency and intensity) shear-rate, resulting in a dramatic decrease in NO-bioavailability. These processes might then cause further consequences, in a vicious cycle, such as a rise in vascular inflammatory markers. In conclusion, knowing the processes involved in vascular deterioration in chronic bedridden individuals, may help in developing successful strategies to add to the standard therapeutic approach. 

## Figures and Tables

**Figure 1 jcm-09-00918-f001:**
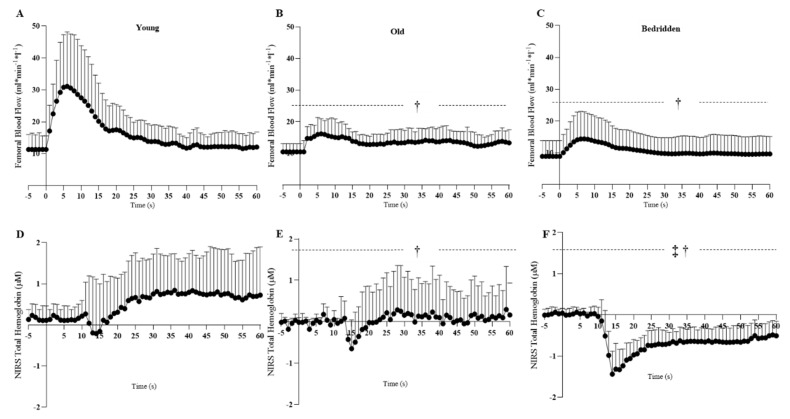
Femoral blood flow and NIRS total hemoglobin during passive limb movement test. Data are presented as mean and standard deviation in the three groups: Young (panel **A**, femoral blood flow normalized for tight volume; panel **D**, NIRS total hemoglobin); Old (panel **B**, femoral blood flow. normalized for tight volume; panel **E** NIRS total hemoglobin); Bedridden (panel **C** femoral blood flow normalized for tight volume; panel **F**, NIRS total hemoglobin). † Between-groups difference versus the Young group (*p* < 0.05) ‡ Between-groups difference versus the Old group (*p* < 0.05).

**Figure 2 jcm-09-00918-f002:**
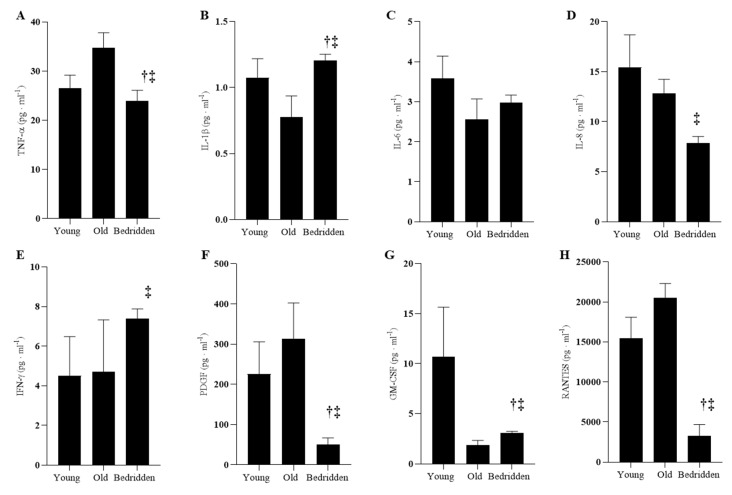
Inflammatory profile and comparison between groups: Young, Old, and Bedridden. Data are presented as mean and standard deviation. Kruskal–Wallis one-way analysis of variance on ranks was used to identify between groups differences. Tumor necrosis factor-α (TNF-α, panel **A**); interleukin-1β (IL-1β, panel **B**); interleukin-6 (IL-6, panel **C**); interleukin-8 (IL-8, panel **D**); interferon-γ (IFN- γ, panel **E**); platelet-derived growth factor (PDGF, panel **F**); granulocyte-macrophage colony stimulating factor (GM-CSF, panel **G**), regulated on activation, normal T cells expressed and secreted (RANTES, panel **H**). † Between-groups difference versus the Young group (*p* < 0.05). ‡ Between-groups difference versus the Old group (*p* < 0.05).

**Figure 3 jcm-09-00918-f003:**
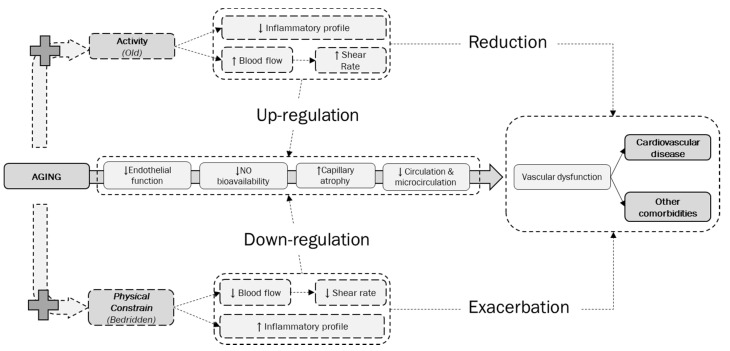
Direct and indirect effects of activity and physical constrain on the physiological aging process. Aging brings with itself several (mal)adaptations at vascular level such as a decrease in endothelial function with a reduction (↓) of Nitric Oxide (NO) bioavailability, followed by an increase (↑) in capillary atrophy leading to a poorer circulation and microcirculation. The worsening of these physiological mechanisms during aging serves as basis for the development of vascular dysfunction and consequently the risk of cardiovascular disease and other comorbidities increases. When the normal aging process is accompanied by physical constraint, with of the absence of movement, as in the case of bedridden individuals, the lack of activity directly impacts blood flow with a consequent reduction in shear-rate together with an altered inflammatory profile. All these events work together in a further downregulation of endothelial function, NO-bioavailability, and capillary atrophy with worst outcomes on circulation and microcirculation. Chronic physical constraint ends in an exacerbation of the vascular dysfunction, leading to a higher risk for cardiovascular disease and other related comorbidities. At the contrary, when the regulator aging process is accompanied by an active lifestyle, as in the case of our active oldest–old, the prolonged and repeated movement serves as a direct stimulus for increasing blood flow and shear-rate, together with a maintenance or amelioration of the inflammatory status, serving as a direct positive stimulus for the endothelial function, NO-bioavailability, maintenance of capillary health, circulation, and microcirculation. Consequently, vascular function is maintained and the risk of developing cardiovascular disease and related comorbidities is reduced.

**Table 1 jcm-09-00918-t001:** Subjects characteristics ^1^.

Characteristic	Young (*n* = 28)	Old (*n* = 36)	Bedridden (*n* = 34)
Age - years	23 ± 3	85 ± 7 ^2^	88 ± 6 ^2^
Female - n (%)	28 (100)	25 (70) ^2^	22 (55) ^2^
Bedridden - years	0	0	3.8 ± 2.3 ^2^^,3^
Weight - kg	60 ± 10	65 ± 12	61 ± 15
Height - m	1.6 ± 0.3	1.6 ± 0.4	1.6 ± 0.6
BMI - kg·m^−2^	22.8 ± 1.8	24.3 ± 4.3	25.2 ± 6.0 ^2^
Tight volume - L	8.4 ±1.5	8.1 ± 1.7	6.7 ± 1.3 ^2^^,3^
Comorbidities			
Number of comorbidities per individual	0	3 ± 1 ^2^	4.0 ± 1 ^2^^,3^
Cardiovascular Disease - n (%)	0	2 (6) ^2^	4 (12) ^2^^,3^
Diabetes - n (%)	0	4 (12) ^2^	3 (9) ^2^
Arthrosis- n (%)	0	3 (9) ^2^	4 (12) ^2^
Pharmacological Treatment			
Number of medications per individual	0	1 ± 0.5 ^2^	4.2 ± 2.2^2^^,3^
Antihypertensive - n (%)	0	4 (12)	8 (24) ^2^^,3^
Cardiological medication – n (%)	0	2 (6)	3 (9) ^3^
Antipsychotics - n (%)	0	0	6 (18) ^2^^,3^
Antidepressant - n (%)	0	0	11 (32) ^2^^,3^
Benzodiazepines – n (%)	0	2 (6) ^2^	7 (20) ^2^^,3^

^1^ Data are presented mean ± SD. Kruskal–Wallis one-way analysis of variance on ranks was used to identify between-group differences. BMI, body mass index. ^2^ Between-group difference versus Young group (*p* < 0.05). ^3^ Between-group difference versus Old group (*p* < 0.05)

**Table 2 jcm-09-00918-t002:** Comparison between groups: Young, Old, and Bedridden ^1^.

Variable	Young (*n* = 28)	Old (*n* = 36)	Bedridden (*n* = 34)
Nitrates - µM	49.5 (38.1–62.4)	44.8 (34.7–56.8)	34.1 (23.8–40.9) ^2,^^3^
Values at rest			
Femoral artery diameter - cm	0.79 (0.73–0.83)	0.78 (0.73–0.83)	0.58 (0.50–0.70) ^2,^^3^
Femoral Blood Flow - mL·min^−1^·L^−1^	43.5 (29.5–51.4)	37.2 (24.1–49.1)	29.7 (20.7–36.1) ^2,^^3^
Femoral artery shear-rate – s^−1^	709 (539–876)	609 (394–701)	452(264–505) ^2,^^3^
Total Hemoglobin - µM ^4^	41.2 (35.9–62.4)	26.4 (14.2–51.8)	22.7 (11.4–25.7) ^2^
sPLM-induced hyperemia			
Blood Flow ∆Peak - mL·min^−1^·L^−1^	65.2 (28.7–114.9)	39.7 (10.5–56.4) ^2^	10.9 (5.8–16.8) ^2,^^3^
Blood Flow AUC - AU	105.8 (57.3–265.5)	17.5 (-2.3–67.7) ^2^	2.3 (−11.5–13.3) ^2^
Total Hemoglobin ∆Peak -µM ^4^	1.5 (1.3–2.5)	0.2 (−0.2–0.9)	−0.5 (−0.6–0.02) ^2^
Total Hemoglobin AUC- AU ^4^	39.0 (25.9–89.1)	−9.4 (−29.6–30.1)	−48.9 (−51.9–−26.5) ^2^
Inflammatory profile			
TNF-α - pg·mL^−1^	33.2 (22.8–39.4)	35.6 (23.5–42.8)	19.7 (18.2–28.1) ^2,^^3^
IL-1β - pg·mL^−1^	0.79 (0.5–0.9)	0.57 (0.4–1.1)	1.17 (1.1–1.4) ^2,^^3^
IL-6 - pg·mL^−1^	2.7 (1.0–4.9)	1.9 (1.9–3.9)	2.7 (2.5–3.1)
IL-8 - pg·mL^−1^	10.1 (6.8–14.5)	13.1 (8.9–15.3)	6.4 (5.7–9.2) ^3^
IFN-γ - pg·mL^−1^	0.82 (0.2–2.7)	3.15 (1.1–9.8)	7.79 (7.2–7.9) ^3^
GM-CSF - pg·mL^−1^	0.92 (0.36–2.36)	1.51 (1.2–2.1)	3.20 (2.9–3.3) ^2,^^3^
PDGF - pg·mL^−1^	160.9 (52.7–331.1)	267.7 (125.1–312.3)	9.7 (7.4–70.2) ^2,^^3^
RANTES - pg·mL^−1^	16,159 (10,162–22,262)	21,612 (16,470–23,046)	33.9 (28.1–35.5) ^2,^^3^

AUC, area under the curve; TNF-α,tumor necrosis factor-α; IL-1β, interleukin-1β;IL-6, interleukin-6; IL-8 interleukin-8; IFN- γ, interferon-γ; PDGF, platelet-derived growth factor; GM-CSF, granulocyte-macrophage colony stimulating factor, RANTES, regulated on activation, normal T cells expressed and secreted. ^1^ Data are presented as median and (25–75 percentile). Kruskal–Wallis one-way analysis of variance on ranks was used to identify between-group differences. ^2^ Between-groups difference versus the Young group (*p* < 0.05) ^3^ Between-groups difference versus the Old group (*p* < 0.05). ^4^ Total hemoglobin was measured by means of near-infrared spectroscopy during the sPLM Test, on a sub-group of 10 individuals for each group.

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
