# Peer review of "The Vascular Side of Chronic Bed Rest: When a Therapeutic Approach Becomes Deleterious"

_jcm, 2020, doi:10.3390/jcm9040918_

Round 1
Reviewer 1 Report
In this clinical study by Pedrinolla Anna et al, they tested a working hypothesis that bedridden individuals would have shown poorer NO-bioavailability, and a poorer systemic circulation and higher inflammatory status compared with active individuals. They assessed NO bioavailability via plasma NO-metabolites and sPLM induced hyperemia, circulation and microcirculation, as well as the inflammatory profile in chronically bedridden oldest-old individuals and compared to a group of age-matched physically active oldest-old and a group of young active adults. They observed that NO-bioavailability, circulation, microcirculation, and inflammatory profile seem to be severely affected by the chronic physical constrain due to years of bed-rest rather than the aging process itself, suggesting a deleterious effect of physical constrain on vascular function, possibly mediated by endothelial dysfunction with reduced NO bioavailability, poor circulation, and systemic inflammation. They further speculated that this might be triggered by the absence of physical activity which does not provide the stimulus, shear-rate, to produce metabolites indispensable for vascular health, such as NO.
It is an interesting study with clinical significance, which may help in the development of effective strategies to add to the standard therapeutic approach.
The study was well designed with interesting observations, the conclusions supported by the data, and the manuscript well written.
Author Response
We thank this reviewer for reading our manuscript and giving us such a good feedback.
Reviewer 2 Report
The manuscript by Pedrinolla et al. investigates the impact of bed-rest on vascular health. It is a well-designed trial, and data is of appropriate quality and well-presented. There are some points that need clarification.
- In the abstract-conclusion authors wrote that inflammatory markers were significantly worst... This description is too subjective. I suggest authors revise this phrase to increase/decrease.
- The bedridden population studied have in average more than 3 years of bed-rest inactivity. This is very interesting as authors excluded most diseases. Can authors add in the discussion or results what were the reasons for this population to be bedridden? Was it just old age?
- Methods line 114, 116 and others. Please revise as authors probably mean thigh when writing tight.
- Methods statistical analysis does not mention any pairing, but results do on line 325. Please clarify. If paired analysis was indeed done, provide details of why and how.
- Table 1 provide interesting results such as number of comorbidities and number of medication. Please comment on what were the most frequent co-morbidities in the bedridden population as well as the most frequent medications. Where these different from the active old population?
- Figure 1 A, B, C indicates that femoral blood flow was similar in old population irrespective of activity. Please comments on this result. Furthermore, this is in agreement with previous publications cited by authors in line 294-295; however authors indicate that results are in disagreement. Please clarify.
- The levels of inflammatory cytokines in blood such as TNFa and IL1b seem significantly higher than in other reports. Can authors compare their results to others and comment on why the differences?
- Authors mention 'inflammaging' however their results suggest that most inflammatory cytokines were lower in bedridden older subjects. Was this unexpected? Please comment on such results.
- Discussion, line 281, authors indicate Figure 2 for Hb results, but Figure 2 has no such data. Please revise.
Author Response
The manuscript by Pedrinolla et al. investigates the impact of bed-rest on vascular health. It is a well-designed trial, and data is of appropriate quality and well-presented. There are some points that need clarification.
We want to thank this reviewer for reading our work and for the important comments received.
Here there is a point-by-point response.
- In the abstract-conclusion authors wrote that inflammatory markers were significantly worst... This description is too subjective. I suggest authors revise this phrase to increase/decrease.
Answer. We thank the reviewer for the suggestion. We understand what this reviewer means but we think we can not change the word used in the abstract. Due to the limited length of the abstract we had to summarize the general trend of the inflammatory markers. Indeed, not all the markers were increased or decreased, for instance: TNF-a, IL-8, PDFG, and RANTES were decreased in Bedridden compared with the other groups; IL-1B, and INF-g were increased in Bedridden compared to Young and Old…and so on. However, depending on the action of the single marker, the general trend of the inflammatory markers highlights a “worst” situation in Bedridden compared to the others. I hope this reviewer can agree with us and accept the terminology we used in the abstract.
- The bedridden population studied have in average more than 3 years of bed-rest inactivity. This is very interesting as authors excluded most diseases. Can authors add in the discussion or results what were the reasons for this population to be bedridden? Was it just old age?
Answer: We thank the reviewer for the suggestion. Indeed, Bedridden participants included in the study were bedridden for an average of 3.8 years. The reason why they were bedridden was mostly related to age and age-related frailty/weakness with consequent motor dysfunction and in some case elevated risk of fall. This clarification was added in the result section, specifically in the subjects’ description.
- Methods line 114, 116 and others. Please revise as authors probably mean thigh when writing tight.
Answer: Thank you for highlighting this error.
- Methods statistical analysis does not mention any pairing, but results do on line 325. Please clarify. If paired analysis was indeed done, provide details of why and how.
Answer: We thank the reviewer for the note. We deleted the word “paired” in the result section since it was a mistake.
- Table 1 provide interesting results such as number of comorbidities and number of medication. Please comment on what were the most frequent co-morbidities in the bedridden population as well as the most frequent medications. Where these different from the active old population?
Answer: We thank the reviewer for the question. We added this information in the result section. “As shown in Table 1, the most frequent co-morbidities in the Bedridden group were cardiovascular disease and arthritis followed by diabetes. Diabetes and arthrosis were common comorbidities in the Old group too. The most frequently used medications were by the Bedridden group were antihypertensive and antidepressant, and benzodiazepines. While in the Old group the most frequently used medications were antihypertensive (Table 1).”
- Figure 1 A, B, C indicates that femoral blood flow was similar in old population irrespective of activity. Please comments on this result. Furthermore, this is in agreement with previous publications cited by authors in line 294-295; however authors indicate that results are in disagreement. Please clarify.
Answer: We thank the reviewer for the note. In the line 294-295 we refer to femoral blood flow at rest, not femoral blood flow during the single passive limb movement test, which is reported in Figure 1. The publications cited in this section shown a reduction in femoral blood flow at rest, that we did not see in our subjects, reason why we indicated that results are in disagreement with previous findings. The increase in femoral blood flow during single passive limb movement is related to Nitric Oxide bioavailability, and this results is explained in the previous subheading, and shown in Table 2 (in the section sPLM - induced hyperemia) as well as supported by Figure 2 (panesl A, B,C).
- The levels of inflammatory cytokines in blood such as TNFa and IL1b seem significantly higher than in other reports. Can authors compare their results to others and comment on why the differences?
Answer: We welcomed the reviewer for the suggestion and we apologize for the typo because all the table and graph measures of cytokine levels should be expressed in pg/ml and not ng/ml. This is the reason why the levels of the measured molecules appear higher than in other publications and reports. This was now corrected in the main manuscript.
- Authors mention 'inflammaging' however their results suggest that most inflammatory cytokines were lower in bedridden older subjects. Was this unexpected? Please comment on such results.
Answer: We found very heterogenous conditions of the serum inflammatory profile in Bedridden individuals, since while IL-8, TNF-α, RANTES and PDGF-b were reduced respect to Old and TNF-α, RANTES and PDGF-b were reduced respect to Young, on the contrary, IL-1β and GM-CSF were increased in the serum of Bedridden respect to both Old and Young, and IFN-γ levels were increased in Bedridden compared to Young. These data suggest that a complex regulation of pro- and anti-inflammatory reactions may occur in Bedridden condition, possibly reflecting a complex modulation of innate and adaptive immune responses. We have now modified and better discussed this issue in the text of the main manuscript.
- Discussion, line 281, authors indicate Figure 2 for Hb results, but Figure 2 has no such data. Please revise.
Answer: We thank the reviewer for the note. We corrected all the reference to the figures since they were mismatched. Now everything should be in place.
